# Surrogate models for the blade element momentum aerodynamic model using non-intrusive Polynomial Chaos Expansions

Rad Haghi[1] and Curran Crawford[1]

[1]Institute for Integrated Energy Systems, University of Victoria, British Columbia, Canada

**Correspondence:** Rad Haghi (rhaghi@uvic.ca)

**Abstract.** In typical industrial practice based on IEC standards, wind turbine simulations are computed in the time domain for each mean wind speed bin using a few number of unsteady wind seeds. Software such as FAST, BLADED or HAWC2 can be used to capture the unsteadiness and uncertainties of the wind in the simulations. The statistics of these aeroelastic simulations output are extracted and used to calculate fatigue and extreme loads on the wind turbine components. The minimum requirement of having six seeds does not guarantee an accurate estimation of the overall statistics. One solution might be running more seeds; however, this will increase the computation cost. Moreover, to move beyond Blade Element Momentum (BEM)-based tools toward vortex/potential flow formulations, a reduction in the computational cost associated with the unsteady flow and uncertainty handling is required. This study illustrates the unsteady wind aerodynamic statistics' stationary character based on the standard turbulence models. This character is shown based on the output of National Renewable Energy Lab (NREL) 5MW reference machine Blade Element Momentum (BEM) simulations. Afterwards, we propose a non-intrusive Polynomial Chaos Expansion (PCE) to build a surrogate model of the loads' statistics, the rotor thrust and torque, at each time step, to estimate the extreme statistics more accurately and efficiently.

## 1 Introduction

The process of calculating loads on wind turbine components is one of the core parts of wind turbine aerodynamic and structural design and optimization. In the last few decades, international organizations have developed different aeroelastic codes such as Fatigue, Aerodynamics, Structures, and Turbulence (FAST) (Jonkman et al., 2005), BLADED (Bossanyi, 2003) and HAWC2 (Larsen and Hansen, 2007) to accurately calculate load time series based on the standardized or site-specific environmental conditions. Engineers and researchers use wind turbine aeroelastic simulation output statistics to calculate extreme and fatigue loads on wind turbine structures and estimate the unsteady power. To take into account the randomness in the unsteady wind, according to IEC standards (IEC 61400-1), the simulation process must use a semi Monte Carlo (MC) method. Therefore, a full simulation set should include a limited number of seeds for generating multiple wind speed time series of $600s$.

In normal practice, for each mean wind speed, at least six different seeded unsteady wind time series are required as the minimum to take into account the uncertainties. This limited number of unsteady simulations does not yield an entirely accurate estimation of the statistics. Gradient-based optimization algorithms may not deal with these inaccuracies well. One option to solve this problem is running more seeds, which will increase the computational cost. The increase in computational cost will

play a more critical role in our decision making if we want to move towards vortex (van Garrel, 2003) and potential flow codes for load calculations, as they require more computation resources inherently. An alternative approach to direct simulation is to use a surrogate model that can provide us with an accurate statistical estimation set based on a limited number of simulations.

The origin of the *surrogate model* lies in Uncertainity Quantification (UQ) analysis (Sudret, 2007). There are many uncertainty quantification implementations in wind energy. More specifically, surrogate models show much potential in wind farm load estimation, wind turbine optimization or reliability analysis. Many researchers have investigated these potentials. For example, Dimitrov et al. (2018); Schröder et al. (2018); van den Bos et al. (2018); Dimitrov (2019) used surrogate models to estimate the loads on a wind turbine based on the stochastic variables gross parameters such as turbulence intensity, mean wind or wind direction. Ashuri et al. (2016); Murcia et al. (2018); Schröder et al. (2020a) used surrogate models for uncertainty propagation through the wind turbine models. More recently, the surrogate models have been used for the wind turbines reliability assessments (Slot et al., 2020; Schröder et al., 2020b). Also, Wang et al. (2020) and Barlas et al. (2021) showed the application of surrogate model in wind turbine optimization. However, very few have looked at building a surrogate model of the aerodynamic model of wind turbine based on the random phases as the input. Fluck and Crawford showed an initial attempt to build a surrogate model based on intrusive Polynomial Chaos Expansion (PCE) on simple lifting line and BEM models (Fluck and Crawford, 2016a, 2018). As they were quickly faced with *curse of dimensionality*, they showed it is possible to reduce the number of random variables in Veers' unsteady wind model significantly. Afterwards, they used this reduced dimension wind model to propagate stochasticity through a simple lifting line (Fluck and Crawford, 2016b) or BEM (Fluck and Crawford, 2018) model. However, with intrusive PCE it is necessary to change the model implementation fundamentally to incorporate the random variables (Sudret, 2007). This might work for a simple model, but when we want to utilize commercially available aeroelastic codes, this will be challenging or even impossible.

This paper's goal is to build a non-intrusive PCE surrogate model of a deterministic aerodynamic model driven by stochastic unsteady wind. This study's implemented aerodynamic model takes wind time series as input and calculates thrust and torque on a NREL 5MW turbine rotor using BEM. The motivation is to build a surrogate model based on a limited amount of simulation data to estimate the statistics of the aerodynamic model output at each time step of the time series quickly. Having this surrogate model at hand helps us explore and experience the opportunities it can provide. This output guides future research in the surrogate model realms for us in the long run. The surrogate model investigation presented is an exploration of the potential benefits and limitations of PCE-based time-domain surrogate models, to help researchers and practitioners develop future surrogate modeling approaches.

As the surrogate models are inherently cheap to run, we take this surrogate model through a Monte Carlo Simulation (MCS) large number of times. The input of these MCS are the samples drawn from the uniform random variables. The unsteady wind generator uses the same random variable distribution to make sure the generated time series will match a Gaussian process (Veers, 1988). This process is presented in Figure 1 schematically. By this method, we can reduce the computational cost and time for the aerodynamic simulation, without compromising the validity of the results. One can interpret this model as a tool to map the input distribution (in this case, an uniform distribution of random seeds-phases) to the output distribution (in this case, distribution of thrust and torque on the rotor).

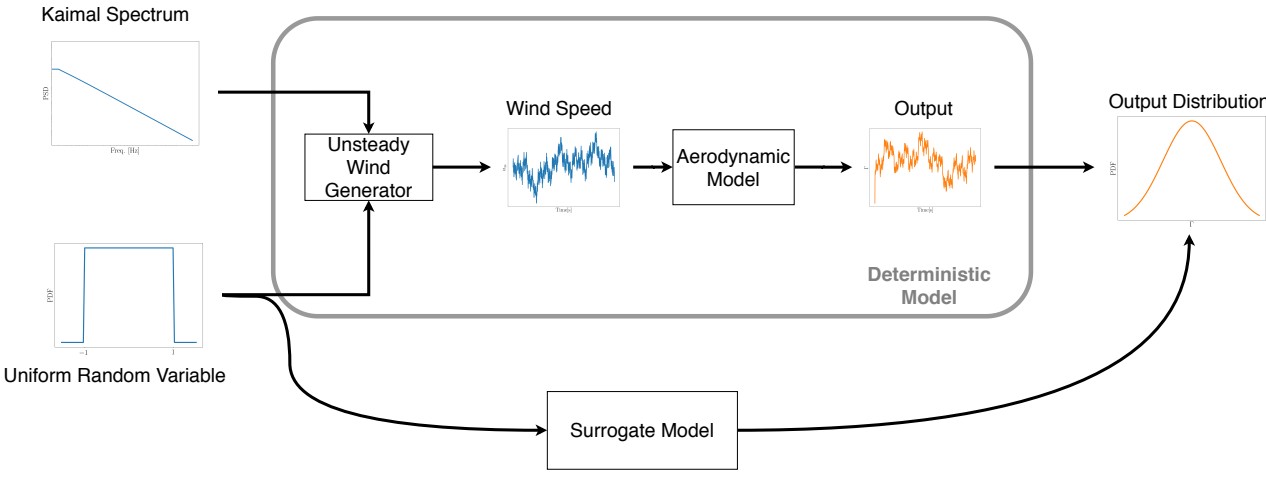

**Figure 1.** The common deterministic process of aerodynamic modeling vs the suggested surrogate model method schematic flow chart.

Fitting a surrogate model at each time step of $600s$ of the aerodynamic output times series, using the random phases (Figure 1), is computationally expensive and redundant to current practice. Therefore, we start by showing that the aerodynamic simulation results based on Veer's reduced model (Fluck and Crawford, 2017) statistically converges. We also show that the unsteady wind aerodynamic process's statistical properties are constant in time (stationary process). Therefore, by keeping the computational effort the same, it is possible to run more simulations while shortening the length of the simulations. Furthermore, more simulations with the same computational effort give us the chance to fit higher degree PCEs, which provides a more accurate estimation of the statistics. We build four different PCEs surrogate models, with four different polynomial degrees (degrees two to five) to pick the best in terms of accuracy and computational cost trade-off. These surrogate models have been used for MCS for a large number of runs (cheaply). The results of the MC runs of the surrogate models are compared with $48000$ unsteady wind aerodynamic simulation results. In this case, the simulation results are the thrust and torque forces induced on the NREL 5MW in an unsteady wind. We compare the results by looking at the thrust and torque distribution from both the deterministic and surrogate models. Finally, we show how the extreme loads extracted from the MCS can converge to the extreme loads from the results from $48000$ unsteady wind aerodynamic simulation results.

This paper is organized as follows. Section 2 describes the unsteady wind generation and aerodynamic model. Sections 3 and 4 explain the and statistical elements and PCE method used in the study. Next, in section 5, we set out the approach to tackle the challenge. Section 6 provides the BEM simulation results, convergence of the sectional statistics the PCE fit on the sectional statistics and the emulations output. At the end of the results section, we discussed the accuracy and efficiency of the surrogate models developed in this study. This paper concludes in Section 7, reiterating the key findings of the study and offering suggestions for fruitful future work in the area of wind turbine surrogates.

## 2 Models

This section provides an overview of the unsteady wind generation basics and aerodynamic model used in this study.

### 2.1 Reduced Veers unsteady wind model

One famous unsteady wind model in the wind turbine community is the Veers model (Veers, 1988). The history of the model goes back to the late 80s' and has a long record in wind turbine load calculation practice. Very briefly, Veers unsteady wind model is inherently an inverse Fourier transformation. The 1-D unsteady wind time series at location $P$ is generated via:

$$u_\infty(t_n) = \sum_m \sqrt{S_m} e^{i(\omega_m t_n + 2\pi \xi_m)} \tag{1}$$

For this inverse Fourier transformation in Eq (1), the frequencies $\omega_m$ and are taken from the Kaimal spectrum (Figure 1). The random phases are based on the independent random variable $\xi_m$ drawn from a uniform distribution over $[0, 1]$. Finally, the amplitude $S_m$ is specified based on the power spectrum at the frequencies $\omega_m$ (Bossanyi et al., 2011).

In load calculation practice, Veers' model for the unsteady wind is the method to generate turbulent boxes, commonly implemented in TurbSim (Jonkman, 2009). The method is briefly explained in Fluck and Crawford (2017) and extensively in Veers (1988). To make the unsteady wind in TurbSim, this method uses a large number of random variables on the order of thousands is required Jonkman (2009). This large number of random variables pushes the surrogate model problem into the *curse of dimensionality* very quickly. Therefore, building a PCE surrogate model is almost impossible. To tackle this issue, Fluck and Crawford (2016b) showed that using only ten uniformly distributed independent random variables with ten frequencies logarithmically sampled from the Kaimal spectrum (Veers, 1988) is enough for building unsteady wind time series. This *Reduced Veers' model* generated unsteady wind that can capture the same level of randomness and probability distribution as the full model. This study used this reduced Veers model to generate unsteady wind time series. This method does not lead to a model that fully replaces high fidelity Turbsim outputs but rather a surrogate model necessity to study trade-offs of various accuracy and assumption aspects.

The randomness in the generated unsteady wind comes from the ten random variables, $\phi_j$, in the $\boldsymbol{\xi}$ vector describing the frequency components' phases $2\pi \xi_m$ in the reduced Veers model in Eq. (1) (Fluck and Crawford, 2017). Based on the Veers method (Veers, 1988) and in TurbSim (Jonkman, 2009), the employed sampling method is a pseudo-Random Number Generator (pRNG) which is the basis of MCS. However, the problem with this way of sampling for MCS is the lack of control over the random variables' domain as it may fill some voids in the domain and may leave some of it empty (Niederreiter, 1992). Therefore, the random domain may not be filled evenly for the same reason. For this study, a low discrepancy Quasi Monte Carlo (QMC) sampling method, namely the Sobol sequence (Sobol', 1967) is used to draw samples from the random variables in this work to calculate the PCE coefficients via the point collocation method. The main reasons to select the Sobol sequence over other sampling methods are the samples' consistency and computational efficiency (Kucherenko et al., 2015). A custom random wind generator based on the reduced Veers model used these samples to generate unsteady wind fields.

## 2.2 Aerodynamic model

The aerodynamic model for this study is a BEM model (Bossanyi et al., 2011) with frozen wake based on the work of Lupton (2019). This non-linear BEM model is used to run simulations on a NREL 5MW Jonkman et al. (2009) rotor to calculate thrust and torque on the rotor. For the simulations, the rotor speed was kept constant based on the mean wind speed of the simulations. Also, the pitch angle was set according to the data provided in Jonkman et al. (2009). There is no controller involved in the simulations. The unsteady wind defined in the previous section is set at 100m hub height and remains the same on the rotor. The Python package for BEM is `bemused` (Lupton, 2019). The NREL 5MW model characteristics and properties are extracted from Jonkman et al. (2009). The model employed in this study for the simulations and surrogate model is essentially equivalent to the NREL 5MW model in any wind turbine aerodynamic code (e.g. FAST, BLADED and HAWC), but nicely formulated in Python. The model and analysis code used in this work has been previously verified against the NREL 5MW full model using BLADED by Lupton (Lupton, 2015).

## 3 Statistical convergence metric

For this study, we want to investigate the null hypothesis that cross-sectional statistics (statistics at each time step) of a combination of a large number of aerodynamic simulation outputs are similar. In other words, we want to investigate if the statistical properties of the output at each time step converge as a function of the number of simulations (stationary process) for the non-linear stochastically autocorrelated system. Figure 2 presents a generic example of distributions (histogram fits) at each time step for a set of realizations of one random process. (The figure shows a schematic plot; therefore, histograms and fitted distributions *do not* represent the plotted trajectories.)

To show that the sectional statistics of a large number of simulations are convergent, we need a metric to quantify the difference between the distributions at each time step. There are different metrics for this purpose (Basu et al., 2011); for this study, we use *Hellinger distance* (Hellinger, 1909) as a metric due to its ease of application and interpretation. The Hellinger distance is the metric to quantify how similar two probability distributions are to each other. The distance is zero if they are the same and is one when the two distributions are disjoint. The Hellinger distance for two *discrete* probability distributions $P$ and $Q$, which have an equal number of bins, can be formulated as:

$$H(P,Q) = \frac{1}{\sqrt{2}} \sqrt{\sum_{i=1}^{k} (\sqrt{p_i} - \sqrt{q_i})^2} \tag{2}$$

In Eq.(2), $p_i$ and $q_i$ are the probabilities for $P$ and $Q$ at every bin. In our case, to make the comparison fair, not only the number of bins are the same, but also the bin width is the same for both $P$ and $Q$. This assures us that there is no artificial distance reduction in the results.

In this study, we use Hellinger Distance to show the cross-sectional statistics changes for a large number of simulations is minimal. Therefore, we can shorten the simulations without losing the accuracy of the sectional statistics. We also use the

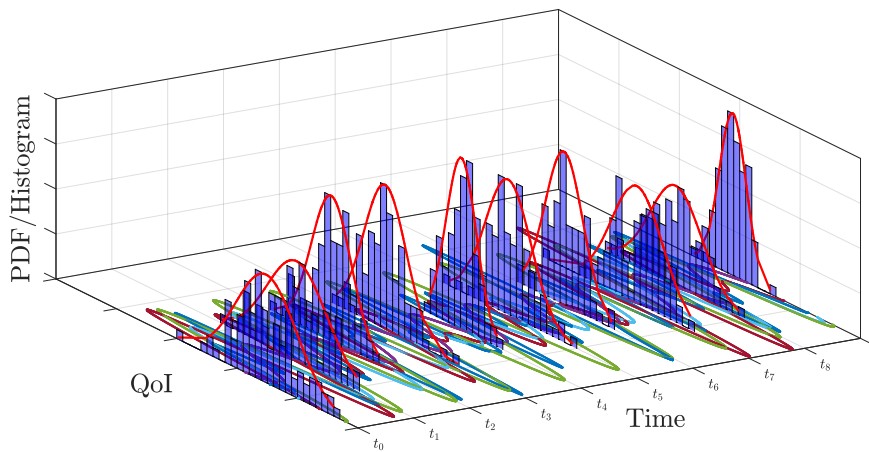

**Figure 2.** A schematic drawing presenting possible distributions at each time step based on a set of time trajectories for a Quantity of Interest (QoI)

Hellinger distance as an error metrics to compare the accuracy of the MCS with the reference case. The reason behind choosing this measure, instead of simply looking at the mean and stand deviation difference, is the distribution of the aerodynamic model output. As the next section we will show, the distribution of the aerodynamic output is a Weibull distribution. Therefore, comparing mean and standard deviation would not provide us the full statistical picture.

## 4  Polynomial Chaos Expansion Fundamentals

Uncertainty propagation of mathematical models has been the subject of many studies in the last thirty years. One method to propagate uncertainty is using models of models, called surrogate models. A surrogate model is a cheap-to-run approximation of the actual model (Kim and Boukouvala, 2019). Among surrogate models, the Polynomial Chaos Expansion (PCE) has gained attention especially after the work of Ghanem and Spanos (2003) and Xiu and Karniadakis (2003). PCE is a method that uses a variable described by its statistical distribution (random variable $\xi$) and projects the model onto a basis of polynomials. This way, the uncertainty can be propagated through the model with a limited number of simulations (Tyson et al., 2015). In other words, PCE is a technique to estimate the response of a mathematical or numerical model based on a series of orthogonal polynomials, which are functions of a random variable $\boldsymbol{\xi}$. The solution is expanded and described in stochastic space spanned by $\boldsymbol{\xi}$ and the associated polynomial basis.

The main reasons to use PCE instead of other surrogate model methods are; a) with minimum computational effort, one can extract statistical moments directly from PCEs; b) PCEs are easy to integrate into deterministic linear and non-linear mathematical models; c) one can build PCE surrogate model by treating the model as a black box (Kaintura et al., 2018; Sudret, 2015) using a non-intrusive formulation.

In order to illustrate the application of PCE surrogates, assume $Y(t_n) = \mathrm{M}(t_n, \boldsymbol{\xi})$ where $t_n$ is time step $n$, and $\boldsymbol{\xi}$ is the random variable vector, $\mathrm{M}(t_n, \boldsymbol{\xi})$ is our deterministic time marching mathematical model and $Y(t_n)$ is the output of the model at time step $n$. Therefore, the stochastic output of the model $Y(t_n, \boldsymbol{\xi})$ can be expanded as:

$$Y(t_n, \boldsymbol{\xi}) = \sum_{i=1}^{\infty} y_i(t_n) \Psi_i(\boldsymbol{\xi}) \tag{3}$$

where $y_i(t_n)$ are PCE coefficients at *each time step* and $\Psi_i(\boldsymbol{\xi})$ is a member of an orthogonal polynomial class. These polynomials are orthogonal with respect to the probability space of random variable $\boldsymbol{\xi}$. The selection of the polynomial type is a function of the probability distribution on the random variable $\boldsymbol{\xi}$. For example, if a random variable $\boldsymbol{\xi}$ has a normal distribution, then a Hermite polynomial is selected (Xiu and Karniadakis, 2002).

The polynomials do not necessarily need to be selected from the specific polynomials family as long as they are orthogonal polynomials. For instance, Fluck and Crawford (2018) showed exponential components worked best for their purposes. As the randomness in this study comes in the form of a uniform distribution for wind frequency component phases $\phi_j$, the surrogate model is based on the Legendre polynomials (Xiu and Karniadakis, 2003). In practice, the PCE summation in Eq (3) is truncated at a reasonably high order $p$. The task of fitting the expansion in Eq (3) is finding the coefficients $y_i(t)$. There are two main approaches to solve this problem:

- the *intrusive* approach where the model is projected on the orthogonal polynomials using a Galerkin projection (Ghanem et al., 2017). This approach requires building a detailed stochastic model from the deterministic model governing equations. The intrusive approach was used by Fluck and Crawford (2016b, 2018) to build a surrogate model on lifting line and BEM models.

- the *non-intrusive* approach allows calculating the PCE coefficients from a series of deterministic model evaluations. This approach considers the model as a black box and does not require any model modification (Sudret, 2007; Eldred et al., 2008). There are two sub-categories to calculate the coefficients, namely *simulation methods* and *quadrature methods* (Sudret, 2007).

The presented work uses the *non-intrusive* approach to calculate the PCE coefficients. In the non-intrusive approach category we primarily used a *simulation method* to calculate the PCE coefficients. In mathematical form, the output of the aerodynamic model $\mathrm{M}_{aero}(t_n, \boldsymbol{\xi})$ at time step $n$ is thrust $Trt(t_n)$ and torque $Trq(t_n)$. Therefore, one can re-write Eq. (3) as:

$$Trt(t_n, \boldsymbol{\xi}) \approx \sum_{i=1}^{m} T_i(t_n)\Psi_i(\boldsymbol{\xi}) \tag{4}$$

$$Trq(t_n, \boldsymbol{\xi}) \approx \sum_{i=1}^{m} \tau_i(t_n)\Psi_i(\boldsymbol{\xi}) \tag{5}$$

where the goal is to calculate the polynomial coefficients $T_i(t_n)$ and $\tau_i(t_n)$ at each time step. This surrogate model's input is the random variable $\boldsymbol{\xi}$ vector used in the reduced Veers model to generate the unsteady wind. The surrogate model's output is the thrust and torque at a specific time step for which the surrogate model is built. The main difference between Eq. (3) and (4)(5) is the approximation with finite polynomial series expansion, as it is not feasible to take into account an infinite number of polynomials. This work's surrogate model is built employing the Python toolbox `chaospy` (Feinberg and Langtangen, 2015). `chaospy` is a numerical tool for uncertainty quantification using different methods, including PCE. For this study, we used the *point collocation* method to calculate the coefficients due to the ease of implementation in the `chaospy` toolbox. This method has been explained well in the literature (see Xiu et al. (2002), Ghanem and Spanos (2003), Sudret (2007)).

## 5   Surrogate Modeling Methodology

In this piece of work, whenever we talk about simulations, we mean aerodynamic simulations in time using the BEM aerodynamic model introduced above. The input of the aerodynamic simulations is what we call wind time series or unsteady wind and is auto-correlated by construct. This wind time series is generated based on the reduced Veers model explained above.

This study starts with running an extensive set of simulations based on the reduced Veers model at $12m/s$ mean wind speed, aerodynamic simulation model and Sobol sampling, explained previously. This wind speed is the rated power wind speed for NREL 5MW. Afterwards, as we have a large number of simulations output in our database, we can show that the thrust and torque statistics with time are not changing significantly. Therefore the statistical process properties at each time step (e.g. mean, standard deviation, etc.) would be significantly similar to other time steps. Knowing the process statistics stays the same in time, we conclude that only building a single surrogate model, i.e., a single time-step or a few ones, would suffice for our purpose. The accuracy of the PCEs depends on the polynomial degree. However, an increase in the polynomial degree pushes the problem further toward the *curse of dimensionality*. The number of required coefficients to build the surrogate model and the required number of simulations are presented in Table 1. Eq (6) presents the formula to calculate the number of PCE coefficients. In Eq. (6), $M$ is the required number of coefficients, $N$ is the number of random variables, and $P$ is the polynomial order.

$$M + 1 = \frac{(N+P)!}{P!N!} \tag{6}$$

The table shows we need a large number of simulations to build an accurate PCE. According to Hosder et al. recommendation, twice the number of simulations $M + 1$ can provide acceptable accuracy for the point collocation method (Hosder et al.). That recommendation is the basis for the number of simulations in this study.

**Table 1.** The number of coefficients and required data points to calculate the coefficients for 10 random variables and using the point collocation method. This number of coefficients should be calculated for every time step. The last column demonstrates the simulation length for the fitted PCEs as explained in Section 6.

| Polynomial Degree | Number of Coefficients + 1 | No of Required simulations | Sims. Length to fit PCEs [s] |
|:---:|:---:|:---:|:---:|
| 1 | 11 | 22 | 163.6 |
| 2 | 66 | 132 | 27.3 |
| 3 | 268 | 536 | 6.7 |
| 4 | 1001 | 2002 | 1.8 |
| 5 | 3003 | 6006 | 0.6 |

We want to limit the computational cost for a single average wind speed to 6 times $600s$ simulations (in total $3600s$) to be competitive with the standard practice in wind turbine aerodynamic simulation. Combining a large number of simulations and $3600s$ cumulative simulations length leads to a large number of short simulations instead of a few long ones. We kept the simulation's cumulative length at 3600 seconds to make this trade-off fair. This means that as the simulations' length decreases, the number of simulations increases. Sobol sampling is the base of the unsteady wind generation and input to the aerodynamic simulation setup. For every set of the required number of simulations in Table 1, the random phases are drawn independently from the rest of the sets. For example, for the second row of Table 1, when we need 132 simulations, 132 unique samples of $\boldsymbol{\xi}$ are drawn from the random domain. These $\boldsymbol{\xi}$ have not been used for other simulation sets. By having a large number of data points at each time step, we built a few surrogate models in time and compared the results with the simulations' reference case. For the sake of accuracy, in this study, we do not build any surrogate model based on 1st-degree polynomials.

Another approach to calculate the coefficients is Gaussian Quadrature (GQ). This method has been extensively explained in the literature e.g. Le Maître and Knio (2010). There are also extensions to GQ referred to as Sparse Gaussian Quadrature (SGQ) methods that seek to reduce the number of simulations required to fit the surrogate (e.g. Smolyak). Our tests show that for a standard GQ method with 10 random variables and polynomial orders 2, 3 and 4 we need 59049, 1048576, 9765625 evaluation points respectively. On the other hand, the Smolyak sparse approach for GQ (Le Maître and Knio, 2010; Smolyak, 1963), will reduce the number of evaluation points drastically. We tested the SGQ method for polynomial orders of 2,3 and 4 with 10 uniform random variables and the Smolyak sparse approach for SGQ. The number of the required evaluations for each polynomial order is 221, 1581 and 8761, respectively. We ran the evaluations for the SGQ method, calculated the weights and built the polynomials. However, the results were not as promising as expected. The results from the SGQ method are shown and briefly discussed in Section 6.4

For a stationary input (Reduced Veers model), the sample statistics of output converges at the rate of $1/\sqrt{n}$, where $n$ is the number of data points (in this case, $48000$ data points at a one-time instance). Consequently, it is possible to *estimate* the statistical parameters of the output distribution by different methods. One possible approach is using the maximum likelihood estimator Rao (2008). A question that then arises is why we go through the complication of building a surrogate model. The research goal we present here is to build a surrogate model of an aerodynamic model, whether the aerodynamic model is simple

or complex, with the model capable of resolving the form of the performance statistics, as an alternative to maximum likelihood estimation methods. We chose an aerodynamic model that is easy to simulate while complex enough to capture the inherent non-linear behaviour. Hence, the specific aerodynamic example model does not compromise the validity of the method we introduce here to later more complex aeroelastic simulations with e.g. FAST or BLADED.

## 6  Results

This section presents the results of our numerical experiments. We start by looking at the Hellinger distance of a large number of aerodynamic simulation output, thrust and torque, and show the distance does not change significantly. Therefore, the sectional statistics are almost the same across time steps. Afterwards, based on that conclusion, we built a number of surrogate models for polynomial orders of 2 to 5 from a limited number of simulations and show the statistics match the reference case. Finally, we show how extracted extreme thrust and torque are comparable with the reference case.

### 6.1  Sectional Statistical Convergence

As mentioned before, in Section 5, we started by running a broad set of reference simulations. For this case, we ran $48000$ simulations for a $12m/s$ wind speed and turbulence intensity of $0.16$. The wind generator code took $48000$ samples from a $10$ dimensional uniform distribution domain based on the quasi-random sampling method. Each sample is a $10$ by $1$ vector of $\boldsymbol{\xi}_j$, and we have $48000$ of them. $48000$ wind speed time histories were generated, and simulations on the aerodynamic models run with a time step of $0.1s$ for $630s$ (in total $6300$ time steps per simulation), with $30s$ transient period. We discarded the transient period for all the processes in this study.

This simulation setup builds a database for the investigation and shows that the process distributions at each time step changes are insignificant. Initially, we started calculating the histogram at each time step with identical binning for all of them. Afterwards, using the Hellinger distance formulation, each histogram's distance to the other histograms (5999 other histograms) was calculated and stored in a matrix. Each row of this matrix shows the histogram difference at one time-step compared to the other ones. Therefore, this is a symmetric matrix with zeros on the diagonal. What is important is the maximum of all of the data in the matrix; in Figure 3a and 3b, we show the max of the Hellinger distance at each time step for the aerodynamic model simulation outputs. The Hellinger distance is a normalized metric, and the distances are shown in the percentage. The plot shows a comparison of all the 18 million possible combinations to calculate the Hellinger distance for each model output. For the simulation outputs (thrust and torque), the distributions' difference does not exceed $2.21\%$. This shows a sound coherence in the statistics at each time step. Therefore, we can conclude that building a surrogate model on a limited number of time steps, or even one time step, is enough, and it is not necessary to create a surrogate model on every time step as predicted by the aerodynamic model form.

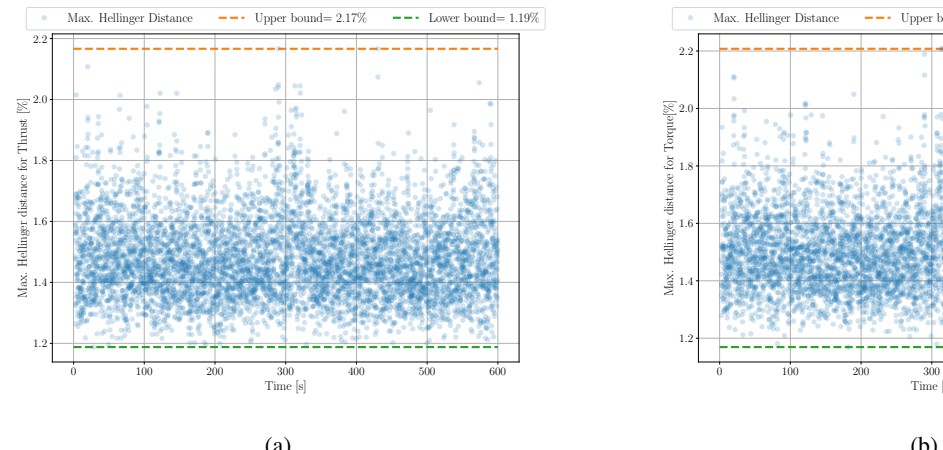

**Figure 3.** Maximum Hellinger distance for thrust and torque at each time step. The upper and lower bounds for the *extreme* of the Hellinger distance are indicated.

## 6.2 PCE Surrogate Model Construction

We use the same simulation setup as explained above for the reference case to run the specified number of simulations in Table 1. These simulations are input for building the surrogate models. The number of samples drawn from the 10 dimensional uniform random space is equal to the number of simulations in Table 1. The employed sampling method is Sobol, as tests show it provides a better convergence for the PCE.

Referring to Figure 3, and the discussion in Section 6.1, the changes in statistical properties at each time step are minimal. Therefore, one surrogate model that can accurately emulate the sectional statistics of the aerodynamic simulations' output would suffice. Knowing this means building surrogate models is more feasible from a computational cost perspective.

As explained in Section 5, we fit surrogate models at every time step of a large set of short simulations instead of a few long ones for increasing polynomials order $P$. The number of simulations is based on the polynomial order as shown in Table 1. The length of the simulations that the surrogate model is built on every time step is the last column in Table 1 to keep the cumulative length of the simulations at $3600s$. Although it was unnecessary, for polynomial degrees 3 to 5, we build surrogate models at every time step of the whole $10s$ worth of simulations to have an acceptable sample size for direct comparison. The `chaospy` (Feinberg and Langtangen, 2015) toolbox was used to perform the task of building these surrogate models.

Figure 4 compares the descriptive statistics (first quartile $Q_1$, second quartile $Q_2$ and third quartile $Q_3$) for both thrust and torque from the reference case with 48K simulation outputs and 48K MCS of the surrogate model build at each time step. The results in Figure 4 show the PCE fits for four polynomial degrees; $P$ on each plot indicate the polynomial degree. As the polynomial degree increases from $P = 2$ to $P = 4$, the fit to the reference case improves, as is expected. However, it seems there is more error in the mean value and quantiles when we move from $P = 4$ to $P = 5$ for both thrust and torque. This

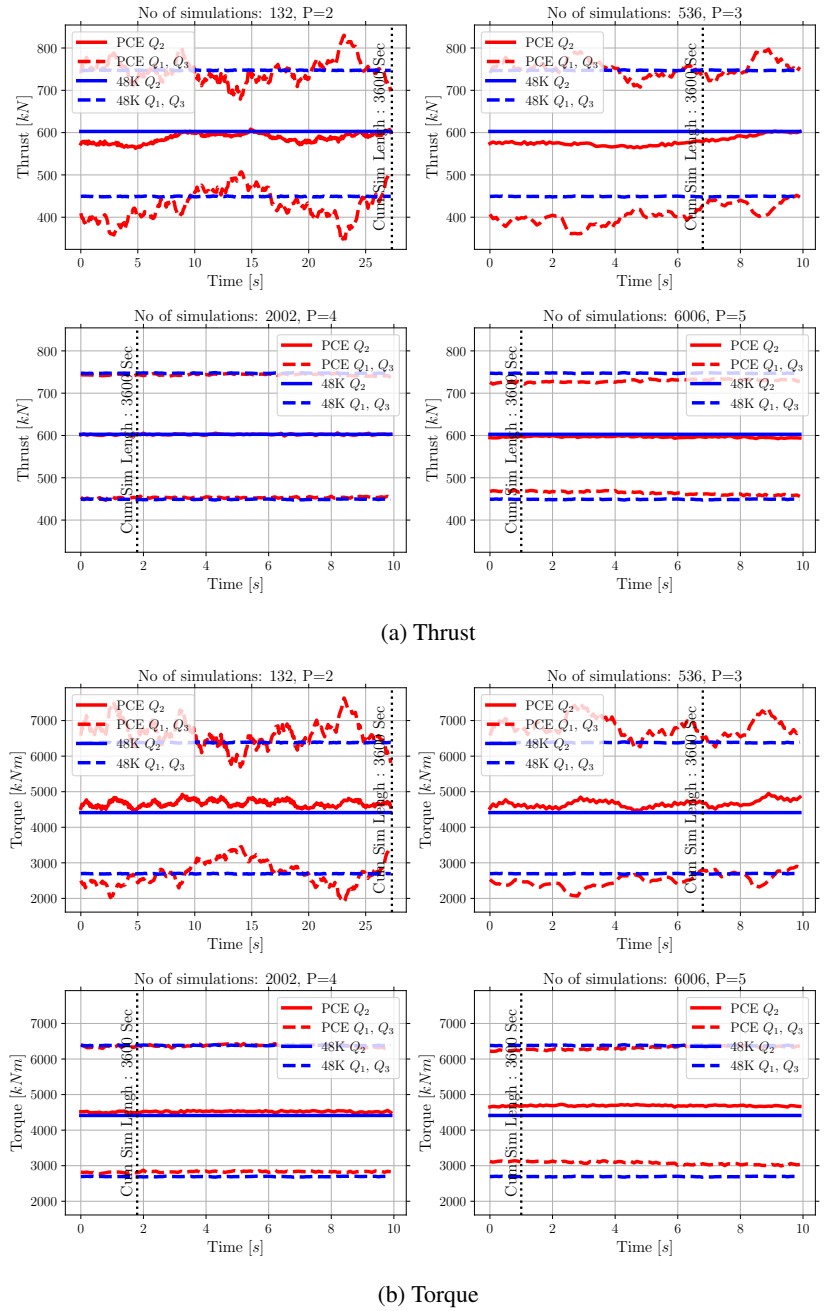

(a) Thrust

(b) Torque

**Figure 4.** The $Q_1$, $Q_2$ and $Q_3$ comparison from the reference case ($48K$ simulations) and extracted values from PCEs for both trust and torque. The number of simulations used to build the PCEs and polynomial degree $P$ are mentioned on the plots. The cumulative data length of $3600s$ sufficient to build surrogates is shown with the vertical line.

increase in the error is explained further in Section 6.3 and Figure 6. We calculate the average difference for the MCS and reference case over time for $Q_1$, $Q_2$ and $Q_3$ from Figure 4 for polynomials $P = 4$ and $P = 5$. This is presented in Table 2.

## 6.3 Surrogate Model MCS

PCE surrogate models were constructed in the previous section. Those surrogates can now be exercised via MCS to quickly provide output sampled statistics without actually running further simulations. We initially ranked the surrogate models constructed for each time-step based on their mean values and standard deviations for each polynomial order separately to choose a single PCE surrogate model for the MCS carried out next. As the surrogate models are constructed as PCEs, mean and standard deviation extraction is a simple step from the PCE coefficients Owen et al. (2017). After ranking the surrogate models, we selected the middle mean surrogate model from the ranked succession for each polynomial order. That provided us 4 surrogate models for thrust and 4 surrogate models for torque, one per polynomial order.

Next, we took the selected surrogate models thorough MCS of the surrogate models $10^6$ times. Essentially, we took random samples from our 10 dimensional random domains $10^6$ times and inserted those in the PCEs (Eq. (4) and Eq. (5)). The MCS outputs are then used to check the surrogate model's accuracy. One can argue that this method is cherry-picking the surrogate models. This argument is valid for the low order ($P = 2, 3$) polynomial surrogate models. However, from Figure 4 we know these polynomial are not accurate regardless of which one we choose. This inaccuracy is more visible in Figure 6. For polynomials of orders 4 and 5, referring to Figure 4, the polynomial selection procedure induces an insignificant effect on the statistics.

To verify the surrogate models' accuracy, we use the Hellinger distance explained in section 5. This time, the Hellinger distance shows the difference between the surrogate model's $10^6$ MCS outputs per polynomial order with the reference case at every time step. This procedure provides a vector of Hellinger distances for each polynomial degree, where the vector's length is the same as the number of time steps in the simulations. As Hellinger distance is sensitive to binning, the bins are identical for each polynomial order surrogate model. The same bins were used to calculate the reference simulations' histograms at each time step. Figure 5 shows the average Hellinger distance changes within a narrow band for each polynomial degree. For the order 4 and 5 polynomilas, we calculate the average of the Hellinger distances over $600s$. The averaged Hellinger distance presented in Table 2 serves as an error metric for the surrogate models.

Afterwards, we compare the histogram of those with one arbitrary time step of the reference case of 48000 simulations. For each polynomial degree, regardless of the reference case time-step location in the time series, the difference between the reference case and the MCS only depends on the polynomial order. In other words, the difference between the MCS result histogram and the reference case histogram was only dependent on the polynomial degree and not the position of the time-step in the time series as expected for a stationary process. Figure 6 compares the histogram of $10^6$ MCS for the middle mean ranked surrogate model to the reference case at one arbitrary time step for four polynomial degrees.

Figure 6 shows how the surrogate models match the simulations output histogram. It is visible that the polynomials order $P = 3$ and $P = 4$ can cover the non-linearity in the results, while the second-order polynomial cannot. Polynomial order $P = 5$ does not work well for both thrust and torque. Although we met the rule of thumb for the number of simulations as mentioned in

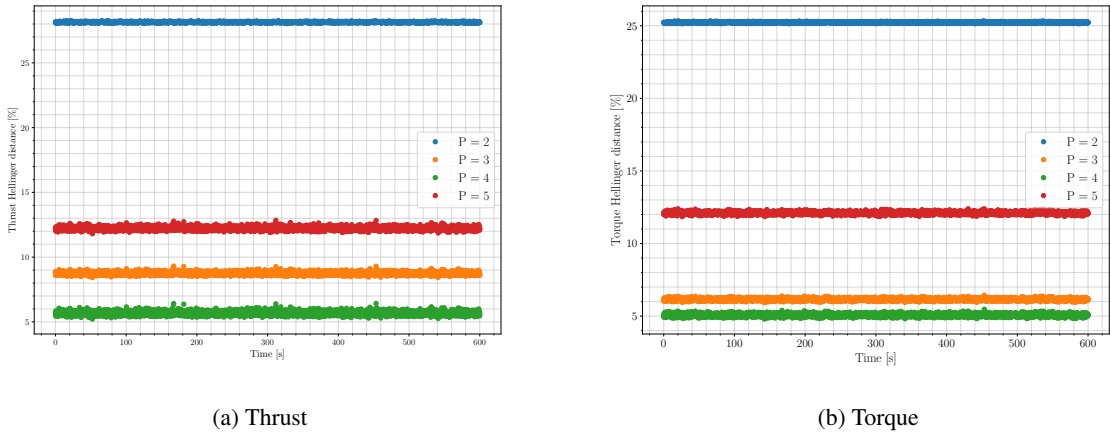

(a) Thrust

(b) Torque

**Figure 5.** Hellinger distance between the different polynomial order surrogate models for one million MCS of the selected thrust and torque surrogates and the reference case simulations $48K$ at every time step

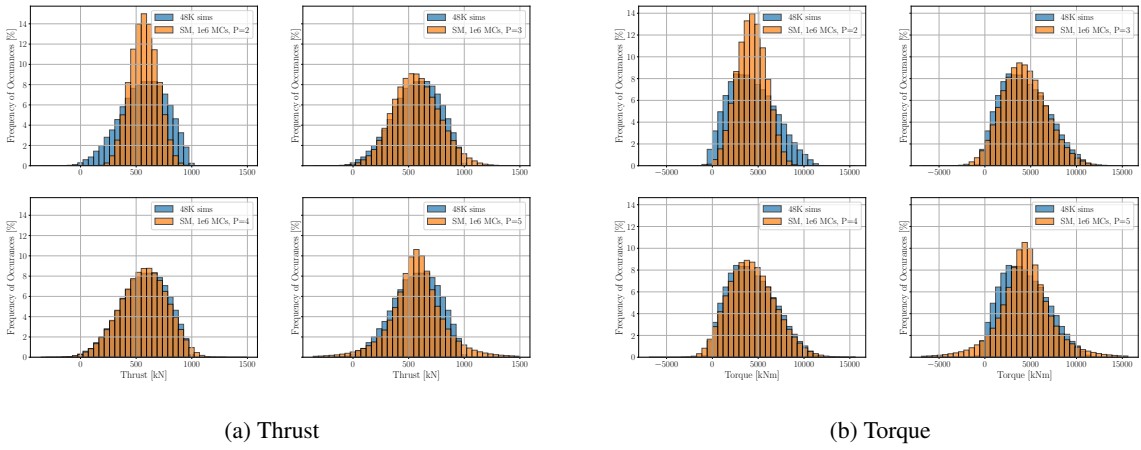

(a) Thrust

(b) Torque

**Figure 6.** Thrust and torque Surrogate models with one million MCS vs the reference case histogram

Table 1, this shows an *under fit* for $P = 5$. This means more simulations are required to make the fit feasible. Both Figures 5 and 6 show that $P = 4$ provides an acceptable accuracy for the surrogate models. Therefore, further tests on the surrogate models with $P = 5$ appear unnecessary. From Figures 5 and 6 we can conclude that the PCE surrogate model with polynomial order $P = 4$ is accurate enough to emulate the aerodynamic model with an acceptable accuracy, while covering its non-linearity.

Another metric to show the accuracy of the surrogate model is Normalized Root Mean Square Error (NRMSE), also known as $L_2$ norm error. The NRMSE is calculated for every time step for the first $10s$ by running MCS for the surrogate model $P = 4$ for the same input as the reference case simulations. This means we use the same samples we took from the $10 - D$ random variables to generate the unsteady wind and then calculate the $48K$ reference case to run the surrogate model MCS. Figure 7

shows the error against time. As expected and visible from Figure 4 the NRMSE is higher for torque and lower for thrust. In both cases, the maximum NRMSE is less than $10\%$. This error is deemed acceptable as the surrogate model aims to provide

overall accurate statistics and not point-to-point accuracy in the estimation and is necessarily a trade-off between speed and accuracy in the intended applications. Recall from Equations (4) and (5) that the PCE method is formulated as an expansion over the space formed by polynomials which are functions of random variables. The simulation method of fitting the PCE coefficients is essentially performing a statistical fit across the summative set of simulation results, rather than optimizing the surrogate fit to a specific simulation. The NRMSE comparison here is therefore perhaps unfair to the intent of the PCE model,

the earlier comparisons of MCS histograms and Hellinger distances more appropriate metrics for the proposed PCE surrogate model approach.

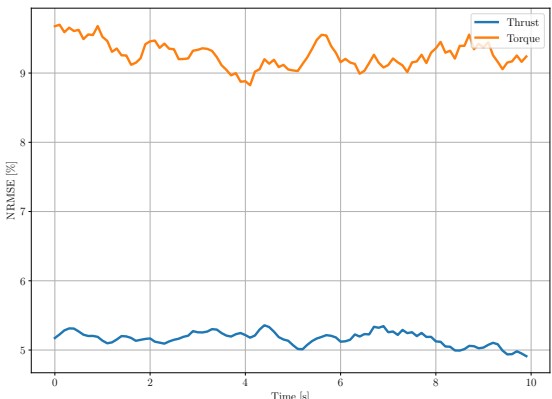

**Figure 7.** Surrogate models $P = 4$ NRMSE for both Thrust and Torque with respect to the reference case

Figures 5, 6 and 7 show the PCE surrogate model has succeeded with the samples from the $10 - D$ uniform distribution and converts them to an approximately Weibull distribution for both thrust and torque. This result highlights the ability of the PCE surrogate in this study to deal with the inherent non-linearity of the combination of unsteady wind generation and aerodynamic

model together.

**Table 2.** The average Hellinger distance for polynomials $P = 4$ and $P = 5$ and the quantile values difference with respect to the reference case.

| Case | No of aerodynamic sims | Avg. Hellinger distance (thrust, torque) [%] | Thrust 1M MCS vs $48K$ sims $Q_1, Q_2$ and $Q_3$ [%] | Torque 1M MCS vs $48K$ sims $Q_1$, $Q_2$ and $Q_3$ [%] |
|---|---|---|---|---|
| Sims for PCE P = 4 | 2002 | 5.6, 5.0 | 0.90, 0.05, 0.50 | 4.94, 2.44, 0.21 |
| Sims for PCE P = 5 | 6006 | 8.7, 6.1 | 3.36, 1.03, 2.42 | 14.12, 6.22 , 1.10 |

## 6.4 SGQ PCE results

As mentioned in Section 5, we also calculated the coefficients for the PCE using the SGQ method for the polynomial orders 2, 3 and 4. We used the same procedure described in the previous sections to run the simulations, build the surrogate models (using `chaospy`), and select the surrogate models. This results in 6 surrogate models (3 for thrust and 3 for torque). We checked the accuracy of the surrogate models in the same manner as explained in the previous section. We ran $10^6$ MCS on the 6 selected surrogate models and compared the histograms of the results with the histogram from one arbitrary time step from the reference case of $48000$ simulations. The results of the investigation are presented in Figure 8.

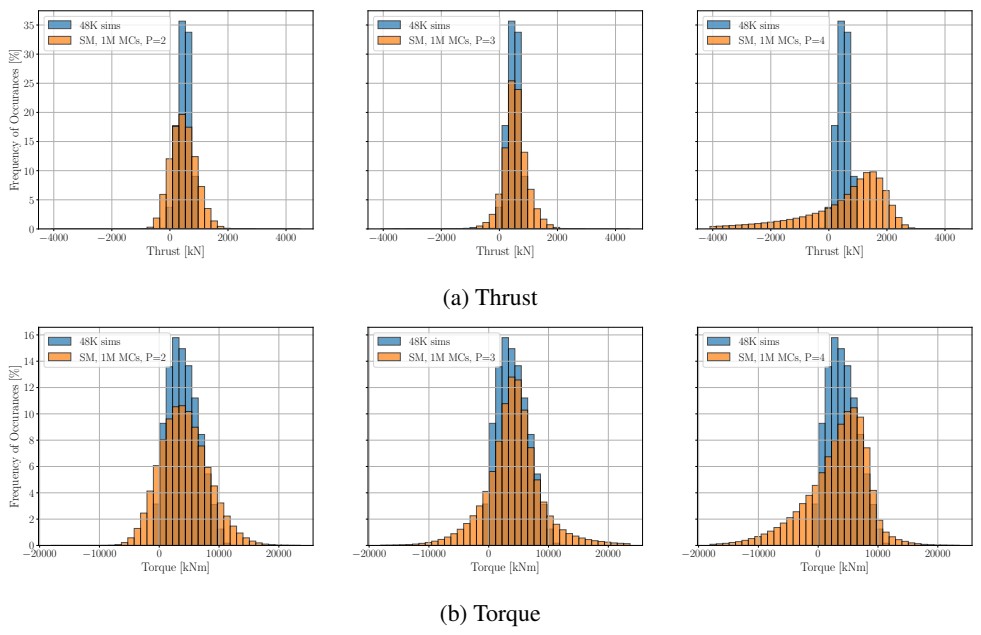

(a) Thrust

(b) Torque

**Figure 8.** Thrust and torque Surrogate models based on SGQ with one million MCS vs the reference case histogram

The results in Figure 8 show the SGQ method is less accurate than the point collocation method used in the previous section. Although the SGQ method requires more evaluation points than the point collocation method, the under-performance of the SGQ method is consistent for all the polynomial orders. As the initial accuracy test for the SGQ method did not provide comparable results with the point collocation method, we did not pursue further investigation of the SGQ method in this study. This finding is in line with literature that shows point collocation typically outperforms the SGQ in accuracy and efficiency (Eldred and Burkardt, 2009).

The Hellinger distance Figures 9a and 9b show the difference between the SGQ surrogate model's $10^6$ MCS outputs per polynomial order with the reference case at every time step. The Hellinger distance is much larger than what we showed in Figure 5 using the point collocation method to build the surrogate models.

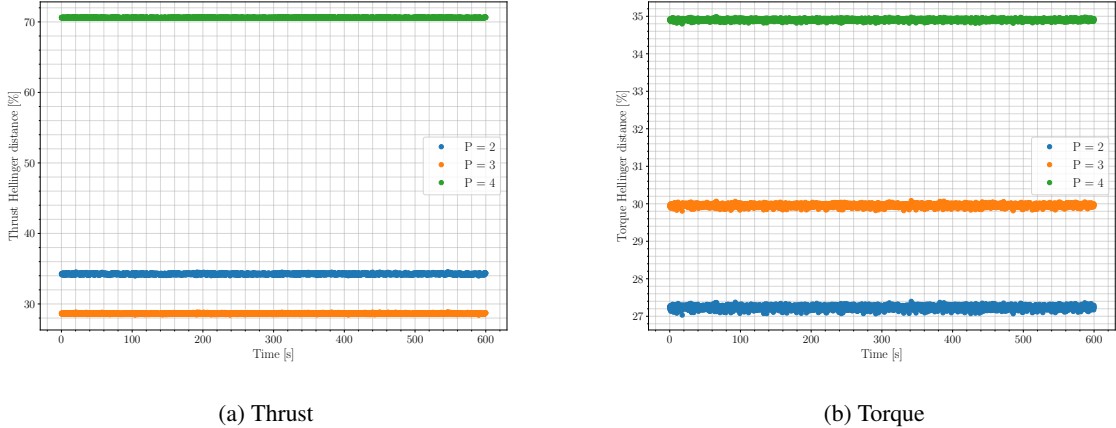

(a) Thrust                            (b) Torque

**Figure 9.** Hellinger distance between the different polynomial order surrogate models for one million MCS of the selected thrust and torque SGQ surrogates and the reference case simulations $48K$ at every time step

## 6.5 Surrogate model efficiency

Building the surrogate models aims to emulate the output of the actual model in an accurate and computationally efficient fashion. To inspect success in this respect, we start with computation time required to run 2002, 6006 and reference case 360 $48K$, $600s$ aerodynamic simulations. The previous section shows that we do not need the $600s$ length simulations to build the surrogate models. Based on what we showed in Section 6.3, the required time to run $1s$ simulations (10-time steps) and building one surrogate model are provided. The computational time in Table 3 includes the unsteady wind generation. Additionally, we record the time required to build one surrogate model. The time to build one surrogate model for both thrust and torque is similar, and the average is provided here. As IEC standards (IEC 61400-1) asks for at least six aerodynamic simulations per 365 average wind speed, we register the time for that set of aerodynamic simulations also. We perform aerodynamic simulations and surrogate model building on Compute Canada WestGrid clusters. The CPU time for the aerodynamic simulations and building the surrogate models is presented in Table 3.

**Table 3.** The computational time required to run aerodynamic simulations, building the surrogate models and the average Hellinger distance

| Case | No of aerodynamic sims | Simulation length | Computation time | Time required build 1 surrogate model |
|---|---|---|---|---|
| Sims for PCE P = 4 | 2002 | 1s | 10s | 1min 2s |
| Sims for PCE P = 5 | 6006 | 1s | 31s | 15min 54s |
| Common practice | 6 | $600s$ | 18.62s | N/A |
| Reference case | 48000 | $600s$ | 41h 55min 48s | N/A |

The computational time to build one surrogate model is long for $P = 5$. This is due to employing the point collocation method to calculate the PCEs coefficients. The point collocation method is inherently a regression method, using least squares to minimize the error (Feinberg and Langtangen, 2015). For a more complex aeroelastic model, the simulation times would be increased, potentially substantially, shifting the balance of computational time from PCE construction toward aeroelastic simulations. Of course, the aeroelastic simulations may be parallelized on available computing infrastructure.

After building the surrogate models, we ran large sets of MCS for the PCE surrogate models with the polynomial order 4 (as it is the most accurate one) and tracked the required time to run the MCS. All the MCS were performed on Compute Canada WestGrid clusters. The computation time for the MCS is shown in Table 4. As the computational time difference between thrust and torque is insignificant, the one which took longer is mentioned here.

**Table 4.** Computational time to run MCS on the surrogate models with polynomial order 4.

| No. MCS simulations | Surrogate model $P = 4$ |
|---------------------|--------------------------|
| 10, 100, 1K | < 0.1s |
| 10 K | < 0.25s |
| 48 K | 0.85s |
| 100 K | 1.70s |
| 1 M | 30.84s |
| 10 M | 5min 51s |
| 100 M | 55min 33s |
| 288 M | 2h 40min 1s |
| 500 M | 4h 46min 36s |

The number of time steps in the reference case is 288 M (6000 time-steps multiplied by 48000 aerodynamic simulations). Therefore, to have a fair comparison we can compare computational time for the reference case in Table 3 with 288 M MCS in Table 4. Adding up the computational time required for the surrogate model input simulations and building the surrogate model, still, the MCS is more efficient by a big margin.

The ease of running MCS provides the ability to have more samples from the random domain. More samples from the random domain cover more of the statistical domain and capture the extreme loads more efficiently than running time marching aerodynamic simulations and extrapolation. Figures 10 and 11 present the comparison between different MCS set sizes and the reference case aerodynamic simulations, maximum, 99th percentile $P_{99}$, 95th percentile $P_{95}$ and 90th percentile $P_{90}$. The maximum load and the percentiles extracted from the $P = 4$ surrogate models run for both thrust and torque are shown in relation to the number of MCS.

According to IEC standards (IEC 61400-1), the maximum load needs to be calculated based on the mean of the max (mean-max) of at least six seeds of unsteady wind aerodynamic simulations per average wind speed. Therefore, we randomly grouped the reference case simulations (48K simulations) into 8000 unique groups to have a fair comparison with the common practice.

Afterwards, the mean of the max per group, and the 90th percentile $P_{99}$, 95th percentile $P_{95}$ and 90th percentile $P_{90}$ of each

group is calculated. These data are presented in Figures 10 and 11 as clouds of dots for both thrust and torque.

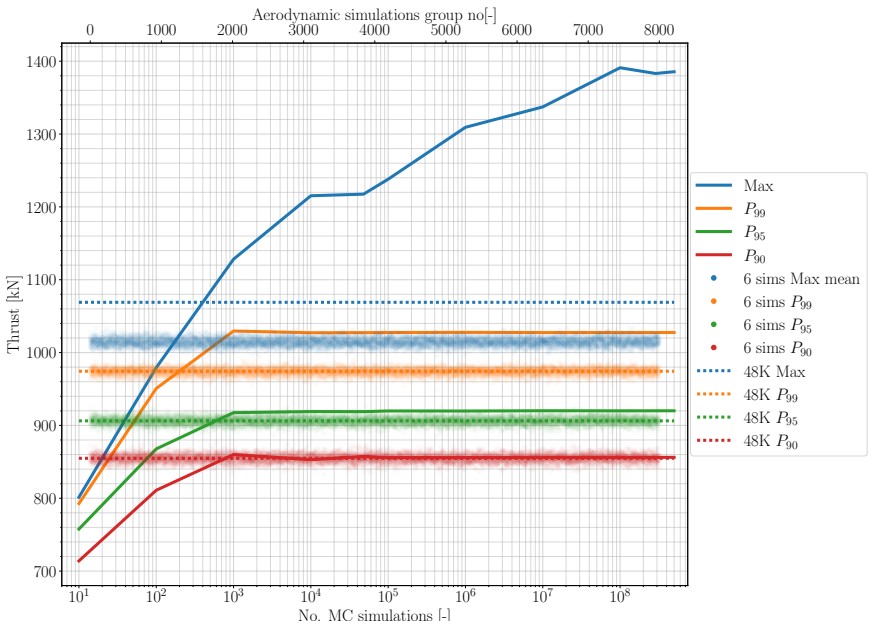

**Figure 10.** Maximum load and percentile comparison between the aerodynamic simulations reference case, the MCS and groups aerodynamic simulations for thrust

    To extract the maximum value that matches the full reference case aerodynamic simulations, we need to run a large number of MCS. Figure 10 and 11 show for the PCE order 4 the maximum thrust and torque from the MCS matches the reference case around almost 1000 MCS. Looking at $P_{99}$, $P_{95}$ and $P_{90}$ for both surrogate models, the convergence happen at around the same

number of MCS. Figures 10 and 11 show after $1K$ or $100K$ MCS the percentiles are close to the reference case. Interestingly, the mean-max output from the grouped aerodynamic simulations has a wide distribution. This shows the inaccuracy of looking at a small number of simulations to calculate the extreme loads. This distribution is smaller for the percentile data; however, it is not comparable with the convergence of MCS outputs. Also, looking at the grouped simulations output, and compared to the standard practice (mean-max) and $P_{99}$, illustrates the conservative design approach of the IEC standards (IEC 61400-1).

Referring to the computational time required to build the surrogate model and then running the MCS, these plots show promising results to extract accurate extreme loads from the surrogate models in a computationally efficient manner. Here again, it is emphasized that the utility of the proposed PCE-based surrogate, with MCS of the constructed surrogate and examination of the statistical loads distributions is the key contribution of the work. The point-to-point accuracy of the model for a single run of the surrogate, as discussed earlier is not the focus of the surrogate, but rather the overall computational costs

and accuracy in spread of loading conditions that is our focus.

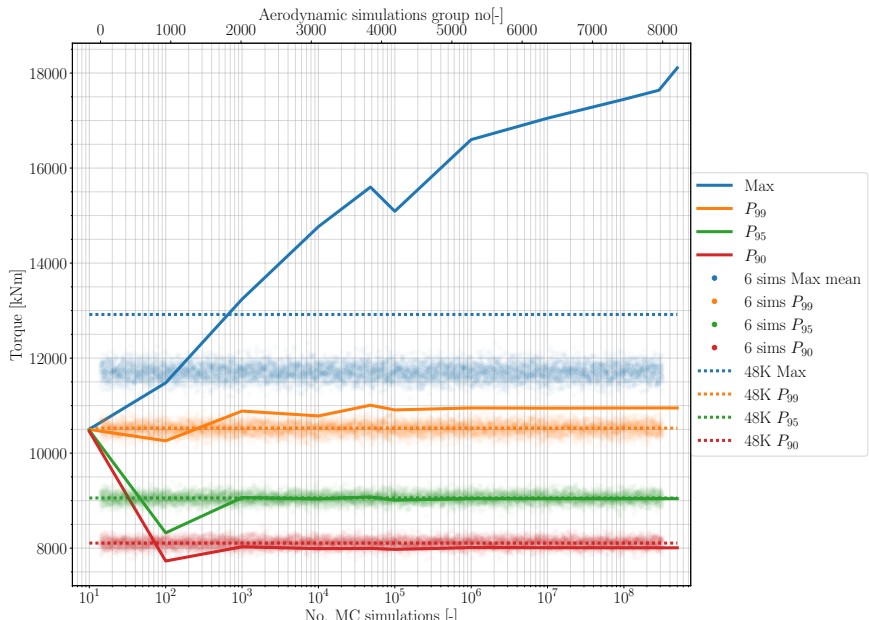

**Figure 11.** Maximum load and percentile comparison between the aerodynamic simulations reference case, the MCS and groups aerodynamic simulations for torque

## 7 Conclusion

This paper's aim is to build a non-intrusive surrogate model of time marching aerodynamic simulations. The form of the surrogate model used in this paper is a PCE. In Section 5, we explained the aerodynamic model used for this study. Then, we briefly described the method that we are using to build the PCEs. One major challenge with the building of the surrogate models is the *curse of dimensionality*, which we tried to tackle by using a reduced Veers model.

We showed how by increasing the number of simulations, the results' statistics converge and do not change in time. As a result, building a few accurate surrogate models, or even one, for a small length of time would suffice for our purpose. In other words, time does not have any meaning in the sectional statistics. Therefore, to build an accurate surrogate model, we can significantly reduce the simulation length while increasing the number of simulations. In the results section, we showed the surrogate model using a fourth-order polynomial built on $2002$ simulations with a length of $2s$ gives us sufficiently accurate results in large MC runs to obtain output statistics. Afterwards, we demonstrate the surrogate model's efficiency by comparing the computational time required to run the aerodynamic simulations and build the surrogate model with the required time for running MCS to have accurate statistics. Also, we show the high percentile values extracted from the MCS match the reference case with a relatively low number of MCS and thus computationally efficient.

The BEM-based aerodynamic model approach is well known in the literature and research. We choose $30s$ transient time for the simulations to ensure they do not include any transition results. As the model is a less complex BEM which is quick

to run, this is not a challenge. However, for future work with actual aeroelastic codes (e.g. FAST), a smart way to deal with initialization time is essential; otherwise, increasing the number of simulations and the model complexity would be very expensive. For example, if the required initialization time is $60s$ (default in FAST), and we want to increase the number of simulations from 6 six hundred second simulations (minimum requirement according to IEC 61400-1) to 6006 two-second simulations, we are not doing any good in terms of computational cost. Aeroelastic and longer wakes will be studied for this challenge and blended or common initialization period will be trialled.

Another challenge is the practical application of this surrogate model. The surrogate model that we build in this study is one or a few time steps, each inherently the same due to the statistical similarity. If we want to build one time series from this surrogate model, we have to sample the 10 dimensional random domain for the number of time steps to have a time series to post-process. For example, suppose we want to have a $600s$ time series of thrust or torque with the time step of $0.1s$ for the aerodynamic model that we developed in this study. In that case, we need to take 6000 samples from the 10 dimensional uniform distribution random domain, and run MCS for each. However, this would provide us with 6000 thrust and torque values, which will miss the auto-correlation, which is inherent in the generated unsteady wind, in the results. This drawback is crucial if we, for instance, want to calculate fatigue loads from the surrogate model. This challenge will require a surrogate form capable of resolving the correlation between time steps. Fluck and Crawford (2018) did this previously for intrusive PCEs of an aerodynamic model, however as mentioned before, that can get very complicated for more advanced models.

Using non-conventional polynomials, such as what Fluck and Crawford (2018) did, might result in a more efficient polynomial that requires fewer number simulations to build the surrogate model. Finally, we want to implement the method on commercial wind turbine simulation packages such as FAST to test the approach in future work. It is important to again note however that the physics model used in the current work is equivalent to FAST, just conveniently formulated in Python for our surrogate model development efforts. The notion of the "reduced Veers model" worked for the aerodynamic simulations we used in this study. However, this reduced model would not be efficient and sufficient to move towards commercial wind turbine packages. Therefore, a new approach to reduce the data in a "turbulence box" Jonkman (2009) spatially and temporally would be necessary. Similar work has been done in Guo and Ganapathysubramanian (2017) and will be explored together with expansions of the methods invested by Fluck and Crawford (2017) using velocity increments across the wind field.

*Code and data availability.* The code and data that support the findings of this study are available from the corresponding author, RH, upon request.

*Author contributions.* RH developed the necessary computer code and wrote the paper in consultation with and under the supervision of CC.

*Competing interests.* The authors confirm there are no competing interests present.

*Acknowledgements.* We greatly acknowledge the funding for this study by the Natural Sciences and Engineering Research Council of Canada (NSERC). This research was enabled in part by support provided by WestGrid (www.westgrid.ca) and Compute Canada Canada (www.computecanada.ca).

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
