# Peer review of "Surrogate models for the blade element momentum aerodynamic model using non-intrusive Polynomial Chaos Expansions"

_Wind Energy Science, 2021_

## Author Response (AR1)

**1 Reviewer one comments**

Thanks for your constructive and concise comments. Here is the reply to your comments:

**1.1 Comment 1**

However, there are issues with two of those three key aspects that would need to be addressed by the authors before releasing this work. First, the curse of dimensionality is not adequately alleviated for the future purposes of this work. Selecting a simpler wind velocity model with a reduced number of inputs does not tackle the problem, just avoids it. When a more complex model is needed, the same technique presented here is not enough to tackle the challenge of dimensionality in PCEs and the work should be revisited entirely. The literature is very rich with mathematical ways of dealing with this issue (low-rank approximations, sparse PCEs etc) and these have not been explored which would be more important for the impact of this work in the field of wind engineering. Furthermore, there are no account of errors for each PCE approximation, for instance for Figure 4, computations of the L2 error norm, not just a qualitative indication.

**1.1.1 Reply**

The authors agree with the comment about the reduced Veers model. We are aware of the challenge and will tackle this in future work. This comment is thoroughly addressed in the new revision. The reduced model showed enough accuracy in Fluck's (2016b) work in covering the variation in the unsteady wind. This reduced Veers model is not a substitute to the high fidelity Turbsim model but a necessity to study the surrogate model at this stage. The sparse PCEs were tested, and the result was not promising in terms of the required data for the Quadrature method. The full gaussian quadrature methods for 10 random variables and polynomial order 4 require 9765625 data points, which means 9765625 simulations. For the sparse setup, 10626 simulations are required. The L2 error for the point-to-point comparison is added to the revised version.

**1.2 Comment 2**

The statistical convergence is assessed using only one metric without giving enough motivation as to why that metric and what errors are committed when using it (how good of an assessment does the Hellinger distance give you? Could it give you a good assessment even in cases that are clearly not good?). It would be interesting to round up that part of the work with another widely used metric, the Kullback-Leibler divergence, used in surrogate modeling when approximating models for Bayesian analysis, for example. Furthermore, when using more complex models, the argument of statistical convergence may not hold due to new underlying physics, so how should we take this?

**1.2.1 Reply**

The KL divergence and Hellinger distance provide similar results, as they are mathematically similar methods. The reasons to use Hellinger distance in this study are its properties and ease of implementation. The Hellinger distance metric has lower (zero) and upper (one) limits. This property makes it a proper metric for this study. This comment is reflected in the new revision of the work. The authors believe the model we implemented here is complex enough to cover the basics for statistical convergence.

**1.3 Comment 3**

An interesting way of enriching this work and properly tackling those three key aspects would be by adopting both the reduced Veers and the full Veers model and make a comparative study with different techniques to build the PCEs and show how you could efficiently used these techniques with a baseline model (reduced Veers) and a more complex one without having to trade some of the physical aspects.

**1.3.1 Reply**

This is indeed an interesting way and can be another paper by itself. However, the full Veers model has, if not hundreds, but thousands of random variables as the input for the unsteady wind generation. This means, considering the curse of dimensionality, the PCEs to build the surrogate models will not be helpful, and it needs other methods to build a surrogate model. We will tackle these ideas in future work. Also, I would like to refer you to the reply to your first comment.

**1.4 Comment 4**

Another concern to be raised is the structure of the paper. It is not well-organized and some sections are a paragraph long. Maybe a better way would be to devote section 2 entirely to the models the authors aim at approximating, with more in-depth discussion. Section 3 could be devoted to the methodology and all the different aspects. Section 4 could then discuss the results.

**1.4.1 Reply**

Thanks for bringing this to our attention. The model in this study is treated as a black box, therefore we do not see any need to go further into the details of the model. We did revise the manuscript to better organize the section and subsections and address your comment.

**1.5 Comment 5**

Technical corrections

Find a list of typos found (non-exhaustive):
Page 4, line 84: dimensionality
Page 4, line 97: properties are extracted
Page 6, line 123: orthogonal polynomials
Page 7, line 173: collocation
Page 9, line 214: Hellinger
Page 12, line 273: we use the Hellinger distance

There are also English inconsistencies throughout the text, with sentences missing verbs etc. The authors should re-read the manuscript carefully.

**1.5.1 Reply**

Thanks a lot for the technical correction. We implemented them and did correct them in the revised manuscript.

**2 Reviewer Two comments**

Thanks a lot for your constructive comments. Here is the response to your comment. In the new revision of the manuscript, we accommodated the majority of your comments.

**2.1 Comment 1**

The key findings of this work should be highlighted at the end of the abstract.

**2.1.1 Reply**

Thanks for the comment. This has been added to the work in the new revision.

**2.2 Comment 2**

1. Introduction

It would be appropriate to add one or two paragraphs to review relevant students on this topic. For example, review relevant studies on developing surrogate models for the calculation of aerodynamic loads on wind turbine rotors. The research gap addressed by this paper should be highlighted. The novelty/contributions of the paper should be highglighted.

**2.2.1 Reply**

Thanks for the comment. The other researchers focused mainly on aero loads, but stochastic variables gross parameters like mean wind or TI. That is different from our purpose. More references of efforts in that other direction, suitable for wind farm dev have been added to the manuscript.

**2.3   Comment 3**

2. Methodology

It would be appropriate to add a flowchart to illustrate the methodology used in this paper.

The accuracy of aerodynamic load calculation is highly dependent on the aerodynamic model. The aerodynamic model used in this study should be elaborated.

It would be appropriate to highlight what the novelty of the methodology proposed by this paper is.

**2.3.1   Reply**

Thanks for your comment. A flowchart is provided in Fig 1. The strength of a surrogate model is the ability to treat the model as a black box. Therefore, the aerodynamic model accuracy is not in the scope of this paper.

**2.4   Comment 4**

3. Results

The aerodynamic model plays a crucial role in the calculation of the aerodynamic loads. Therefore, it would be appropriate to perform case studies to validate the simplified aerodynamic model used in this studay. For example, perform a case study to compare the aerodynamic torque and thrust obtained from the simplied aerodynamic model against the results obtained from NREL FAST code.

For the surrogate model, it would be appropriate to list all the independent variables (i.e. input variables) and dependent vairables (i.e. output variables).

More case studies should be performed to validate the surrogate model. For example, the results obtained from the surrogate model should be compared against the results obtained from the direct simulation using aerodynamic model. The R squared value should be presented.

**2.4.1   Reply**

Thanks for your comment and concern. We address this thoroughly in the revised manuscript. The simplified BEM model is verified against full NREL 5MW model in Bladed by Lupton in his PhD thesis.

This study aims to look at the statistics of the loads. The statistics of the surrogate model outputs are compared with the output of the aerodynamic model outputs in figure 5 by using the Hellinger distance. In this case, the reference is the 48K aerodynamic simulations and the plot shows how the output of 1 million MCS of the surrogate models differs from the actual simulations. Also, an L2 error for point-to-point comparison between the aerodynamic model and surrogate model is added to the revised manuscript to address your concerns.

---

## Referee Report (RR1)

**Reviewer comment for the WES paper**

**"Surrogate models for the blade element momentum aerodynamic model using non-intrusive Polynomial Chaos Expansions"**

**by Rad Haghi and Curran Crawford**

**Specific comments that remain after the revision**

**Referee comment #1**

There are issues with two of those three key aspects that would need to be addressed by the authors before releasing this work. First, the curse of dimensionality is not adequately alleviated for the future purposes of this work. Selecting a simpler wind velocity model with a reduced number of inputs does not tackle the problem, just avoids it. When a more complex model is needed, the same technique presented here is not enough to tackle the challenge of dimensionality in PCEs and the work should be revisited entirely. The literature is very rich with mathematical ways of dealing with this issue (low-rank approximations, sparse PCEs etc) and these have not been explored which would be more important for the impact of this work in the field of wind engineering. Furthermore, there are no account of errors for each PCE approximation, for instance for Figure 4, computations of the L2 error norm, not just a qualitative indication.

**Authors' response**

 The authors agree with the comment about the reduced Veers model. We are aware of the challenge and will tackle this in future work. This comment is thoroughly addressed in the new revision. The reduced model showed enough accuracy in Fluck's (2016b) work in covering the variation in the unsteady wind. This reduced Veers model is not a substitute to the high fidelity Turbsim model but a necessity to study the surrogate model at this stage. The sparse PCEs were tested, and the result was not promising in terms of the required data for the Quadrature method. The full gaussian quadrature methods for 10 random variables and polynomial order 4 require 9765625 data points, which means 9765625 simulations. For the sparse setup, 10626 simulations are required. The L2 error for the point-to-point comparison is added to the revised version.

**Referee comment to authors' response**

It's still not clear how building a surrogate for a simpler model helps with a more complex model. Surrogate modeling is highly dependent on the problem at hand (e.g. behavior of the function to approximate, discontinuities, non-linearities, correlation, etc). If the problem at hand completely changes (more complex model) then the whole methodology here described will have to be re-assessed entirely. The authors should detail what exactly is the usefulness of approximating this simpler model, what is this "necessity" you mention? If the purpose is to guide future research for

more complex models, as repeatedly mentioned throughout the manuscript, then a wider exploration of options and discussion on how well they adapt to your problem should be priority.

You only mention the number of evaluations required but for what level of fidelity/error? Generally, an L2 error norm vs points used for surrogate building plot is used for the rigorous comparison. How many points you need for what level of error, etc. Rigorous discussion is missing for proper justification of the approach and not just comparison based on number of evaluations. You may be happy with the error you get with the point-collocation method but the error with the sparse quadrature approximation might be smaller (even if you don't want to pay the price of the extra simulations), in which case the comparison is not fair, and again, all of this is with the foresight of going to more complex models in the future for which this aspect might be important. In this regard, the L2 error norm of Figure 7 seems rather large. A good surrogate model should not exceed 1% error. The argument made by the authors who claim to care about the statistics should be revised by computing the same metric on the statistics instead. Error on the standard deviations and means, for instance. Is that error acceptable and why?

**Referee comment #4**

Another concern to be raised is the structure of the paper. It is not well-organized and some sections are a paragraph long. Maybe a better way would be to devote section 2 entirely to the models the authors aim at approximating, with more in-depth discussion. Section 3 could be devoted to the methodology and all the different aspects. Section 4 could then discuss the results.

**Authors' response**

Thanks for bringing this to our attention. The model in this study is treated as a black box, therefore we do not see any need to go further into the details of the model. We did revise the manuscript to better organize the section and subsections and address your comment.

**Referee comment to authors' response**

Fine with referencing the model and treating it in black box fashion but I don't see changes in the paper structure. The same concerns remain. For instance, Section 2. mixes the physical model, surrogate model, statistical convergence, etc. It is confusing. Section 2.5 is one paragraph long.

---

## Author Response (AR2)

Thank you for constructive and concise comments. Here is the reply to your comments:

**1 Comment 1**

It's still not clear how building a surrogate for a simpler model helps with a more complex model. Surrogate modeling is highly dependent on the problem at hand (e.g. behavior of the function to approximate, discontinuities, non-linearities, correlation, etc). If the problem at hand completely changes (more complex model) then the whole methodology here described will have to be re-assessed entirely. The authors should detail what exactly is the usefulness of approximating this simpler model, what is this "necessity" you mention? If the purpose is to guide future research for more complex models, as repeatedly mentioned throughout the manuscript, then a wider exploration of options and discussion on how well they adapt to your problem should be priority. You only mention the number of evaluations required but for what level of fidelity/error? Generally, an L2 error norm vs points used for surrogate building plot is used for the rigorous comparison. How many points you need for what level of error, etc. Rigorous discussion is missing for proper justification of the approach and not just comparison based on number of evaluations. You may be happy with the error you get with the point-collocation method but the error with the sparse quadrature approximation might be smaller (even if you don't want to pay the price of the extra simulations), in which case the comparison is not fair, and again, all of this is with the foresight of going to more complex models in the future for which this aspect might be important. In this regard, the L2 error norm of Figure 7 seems rather large. A good surrogate model should not exceed 1% error. The argument made by the authors who claim to care about the statistics should be revised by computing the same metric on the statistics instead. Error on the standard deviations and means, for instance. Is that error acceptable and why?

**1.1 Reply**

Thank you for your comment.

- As the reviewer mentioned, the main goal is to guide future research, and more clarification is added to the manuscript.

- The mentioned necessity is not about the model simplicity but the reduced Veers model. To make the surrogate model work, using the reduced Veers model is necessary. The used model is not simple. The model is less complex than the model used in the commercial wind turbine simulation packages.

- We tested the Sparse Gaussian Quadrature (SGQ) method to build the surrogate model. The error of the SGQ was much higher than the point-collocation method. We added a section to present the results and errors from the SGQ method.

- We used Hellinger distance as a statistical error metric in this study. It means we used Hellinger distance to show the convergence of the reference case and the statistical error of the surrogate model. We clarified the applications of Hellinger distance in the manuscript. As the output distribution is Weibull, we calculate the error for $Q_1$, $Q_2$ and $Q_3$ for the reference case from Figure 4.

- While working on the revised version, we found an error in the code to calculate Hellinger distance. All the plots that show Hellinger distance is updated accordingly.

**2 Comment 2**

Fine with referencing the model and treating it in black box fashion but I don't see changes in the paper structure. The same concerns remain. For instance, Section 2. mixes the physical model, surrogate model, statistical convergence, etc. It is confusing. Section 2.5 is one paragraph long.

**2.1 Reply**

Thanks for your comment. We updated the paper structure to tackle this issue. We restructured the paper into six sections without any small subsections to make the manuscript more clear.